# Aneuploidy as a cause of impaired chromatin silencing and mating-type specification in budding yeast

Wahid A Mulla[1,2], Chris W Seidel[3], Jin Zhu[1], Hung-Ji Tsai[1], Sarah E Smith[3], Pushpendra Singh[1], William D Bradford[3], Scott McCroskey[3], Anjali R Nelliat[1,4], Juliana Conkright[3], Allison Peak[3], Kathryn E Malanowski[3], Anoja G Perera[3], Rong Li[1,4]*

[1]Department of Cell Biology, Center for Cell Dynamics, Johns Hopkins University School of Medicine, Baltimore, United States; [2]Department of Medicine, McKusick-Nathans Institute of Genetic Medicine, Johns Hopkins University School of Medicine, Baltimore, United States; [3]Stowers Institute for Medical Research, Missouri, United States; [4]Department of Chemical and Biomolecular Engineering, Whiting School of Engineering, Johns Hopkins University, Baltimore, United States

**Abstract** Aneuploidy and epigenetic alterations have long been associated with carcinogenesis, but it was unknown whether aneuploidy could disrupt the epigenetic states required for cellular differentiation. In this study, we found that ~3% of random aneuploid karyotypes in yeast disrupt the stable inheritance of silenced chromatin during cell proliferation. Karyotype analysis revealed that this phenotype was significantly correlated with gains of chromosomes III and X. Chromosome X disomy alone was sufficient to disrupt chromatin silencing and yeast mating-type identity as indicated by a lack of growth response to pheromone. The silencing defect was not limited to cryptic mating type loci and was associated with broad changes in histone modifications and chromatin localization of Sir2 histone deacetylase. The chromatin-silencing defect of disome X can be partially recapitulated by an extra copy of several genes on chromosome X. These results suggest that aneuploidy can directly cause epigenetic instability and disrupt cellular differentiation.
DOI: https://doi.org/10.7554/eLife.27991.001

*For correspondence:
rong@jhu.edu

Competing interests: The authors declare that no competing interests exist.

## Introduction

The histone modification landscape and the associated open or closed (silenced) chromatin conformations regulate access to the genetic information by the transcriptional machinery and provide a mechanism for the establishment and maintenance of stable epigenetic states in well-differentiated cells and tissues (*Jaenisch and Bird, 2003*). Alterations in epigenetic modifications have been recognized as a key step in the initiation and progression of cancer whereby quiescent or slowly-dividing somatic cells escape their normal differentiated state and undergo precocious proliferation. (*Feinberg et al., 2006*; *Timp and Feinberg, 2013*; *Morgan and Shilatifard, 2015*). However, the mechanisms underlying changes in the epigenetic-state associated with neoplastic transformation have not been fully elucidated.

Cancer progression is also associated with a wide range of genetic abnormalities, from mutations of single genes to structural or copy number alterations on the chromosomal level. Aneuploidy, an unbalanced genomic state in which the number of chromosomes deviates from a multiple of the haploid complement, is found in over 90% of human solid tumors and 50% of hematopoietic malignancies (*Mitelman et al., 2012*). Although the association of aneuploidy with cancer was noted more than a century ago, its contribution to cancer progression has only been actively explored in

recent years (*Boveri, 1914*; *Holland and Cleveland, 2012*). Aneuploidy is correlated with complex patterns of altered gene transcription (*Upender et al., 2004*; *Ried et al., 2012*), but its potential impact on the epigenetic state of cancer cells remains unclear due to the co-existence of the other genetic alterations in highly complex and unstable cancer genomes.

Studies in unicellular eukaryotes, such as the budding yeast *Saccharomyces cerevisiae*, have provided valuable insights into the effects of aneuploidy on gene expression and corresponding cellular phenotypes because diverse aneuploid strains that differ only in chromosome stoichiometry, but not in DNA sequence, can be readily generated (*Pfau and Amon, 2012*; *Mulla et al., 2014*). In yeast, chromosome copy number variation leads to scaled changes in the transcriptome and proteome for most of the genes carried on the aneuploid chromosome, as well as expression level changes that vary significantly more than the scaled amount for 5–10% of total genes distributed throughout the genome (*Torres et al., 2007*; *Rancati et al., 2008*; *Pavelka et al., 2010*; *Sheltzer et al., 2012*). The widespread but mostly moderate gene expression changes caused by aneuploidy lead to quantitative alterations in cell growth under a wide range of environmental conditions. However, the existing yeast studies have not addressed whether aneuploidy has the potential to alter the stable epigenetic states correlated with cellular differentiation. This is in part because yeast cells lack complex developmental fates and yeast genome comprises mostly open chromatin accessible to the transcriptional machinery (*Millar and Grunstein, 2006*).

Yeast cells, however, do have a few well-established regions of silenced chromatin, including the cryptic mating type loci *HML* and *HMR* on chromosome III, the *rDNA* repeats on chromosome XII, and subtelomeric regions (*Bühler and Gasser, 2009*). In particular, chromatin silencing at *HML* and *HMR* is critical for the specification of the sexual identity of yeast, in the form of *a* or $\alpha$ mating type, which is stably inherited from generation to generation. The underlying epigenetic mechanism of mating type specification depends on the recruitment of the Sir2 NAD-dependent histone deacetylase to *HM* loci through interactions with other Sir proteins (Sir1, 3, and 4) and several other accessory factors (*Liou et al., 2005*; *Kueng et al., 2013*; *Behrouzi et al., 2016*). Spreading of the Sir protein complex across this region of DNA leads to hypoacetylated histones and establishes stably silenced chromatin (*Rusche et al., 2003*).

In this study, we took advantage of the genetic tools available in yeast and used *HML* silencing as the primary readout to test whether aneuploidy can affect cell identity by disrupting heterochromatin chromatin assembly and maintenance. By inducing meiosis in triploid cells, we generated thousands of aneuploid colonies and screened them using an imaging-based assay to determine the frequency at which aneuploid karyotypes disrupted transcriptional silencing at *HML*. Using a battery of genomic, transcriptomic and cell biological analyses, we investigated the mechanisms by which aneuploidy caused defects in chromatin silencing and epigenetic inheritance.

## Results

### Diverse aneuploid karyotypes can cause chromatin desilencing

To investigate whether and at what frequency random chromosome stoichiometries could disrupt chromatin silencing, we started with a haploid yeast strain containing a yellow fluorescent protein (YFP) with a nuclear localization sequence under the *URA3* promoter, which was inserted into the silent *HML* locus. This *HML::YFP* reporter was shown previously to respond to transcriptional silencing in a Sir2, and 3-dependent manner, like the genes that normally reside at the silent mating type loci (*Xu et al., 2006*). We converted the haploid strain carrying *HML::YFP* to a fully isogenic and homozygous triploid strain by cycles of mating-type switching and mating (*Figure 1—figure supplement 1A*) as previously described (*Pavelka et al., 2010*). The resulting triploid strain, which exhibited complete silencing at the *HML* locus as indicated by the lack of YFP fluorescence (*Figure 1B*), were then sporulated and viable meiotic progenies were isolated through tetrad dissection. Previous studies showed that ~100% of the resulting colonies were aneuploid with random combinations of chromosome numbers, due to the segregation of 3 sets of homologous chromosomes during meiosis (*Campbell et al., 1981*; *Pavelka et al., 2010*; *St Charles et al., 2010*). Using fluorescence microscopy, we examined and identified individual colonies with defects in the silencing of YFP at the *HML* locus. Roughly 3% (98 out of 3418) of viable aneuploid spore colonies exhibited varying degrees of silencing defects. In contrast, we did not observe silencing defects in haploid meiotic progenies

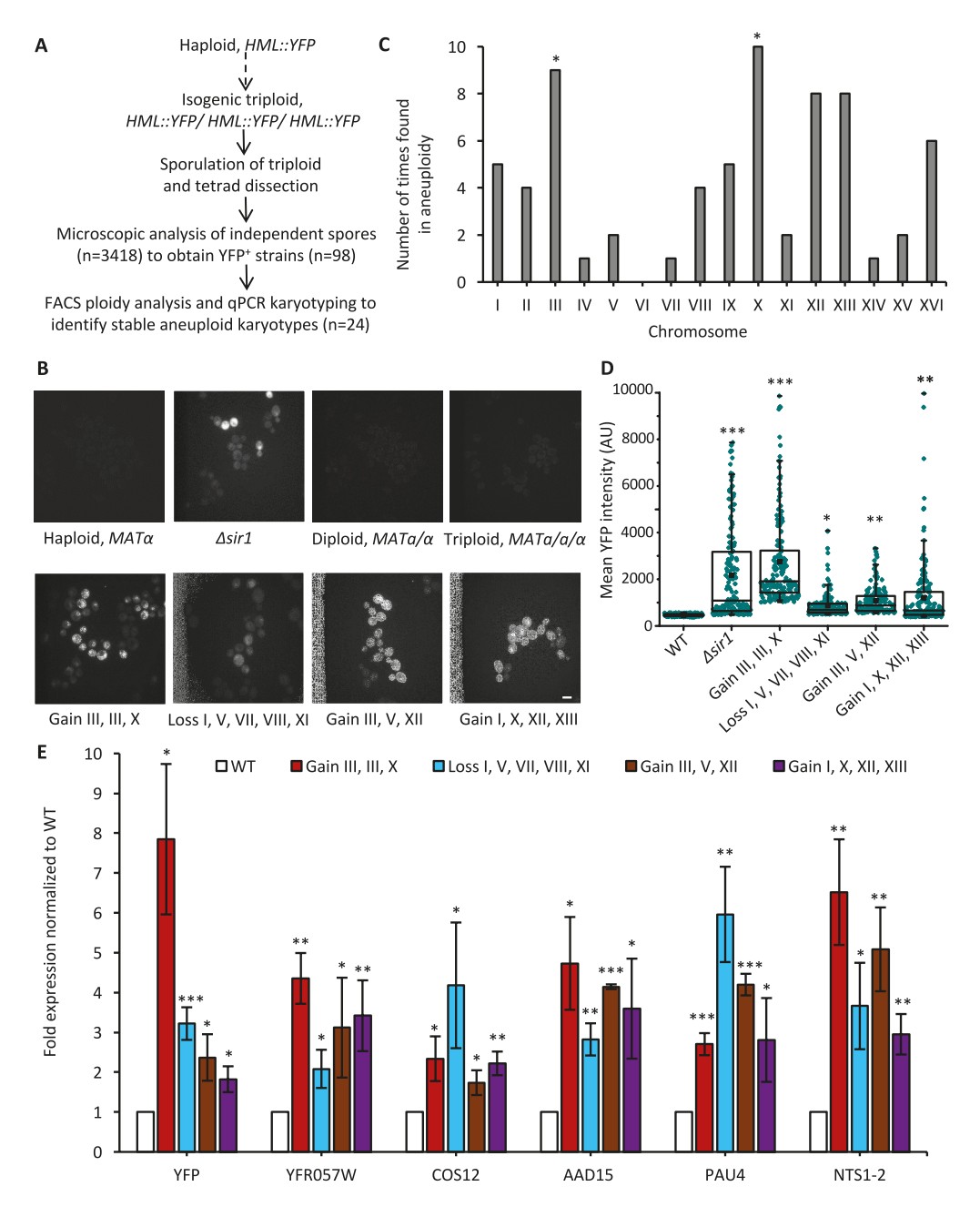

**Figure 1.** Aneuploid yeast strains show defective silencing at *HML*, subtelomeric, and *rDNA* chromatin regions. (**A**) The design of a microscopy-based screen to isolate karyotypically stable aneuploid strains, generated by inducing triploid meiosis, that exhibit defective silencing of the *HML* locus. (**B**) Representative fluorescence images show *HML::YFP* reporter expression in euploid and aneuploid cells of various karyotypes, as indicated. YFP expression from the *HML* locus is not detectable in the parental haploid, diploid and triploid strains; YFP fluorescence is heterogeneous within Δ*sir1* and aneuploid cell populations suggests defective silencing at the *HML* locus. Scale bar, 4 μm. (**C**) The bar plot shows the number of times each of the sixteen yeast chromosomes was found to be aneuploid (chromosome number different from the basal ploidy) in 24 strains with defective silencing. Aneuploidies of Chr III and Chr X are significantly overrepresented in strains with defective silencing compared with other 38 stable aneuploids isolated by the same method (*Pavelka et al., 2010*). *p<0.05 for Chr III and Chr X; p=0.09 for Chr XII calculated using an exact binomial test. (**D**) The box plot shows mean YFP intensities, determined by microscopy as in *Figure 1B*, for 175 individual cells per strain. The karyotype of each aneuploid strain is indicated; WT and Δ*sir1* cells are haploid. The box spans the first through third quartile values, the line inside each box indicates the median, the solid black square designates the mean, and the whiskers mark the 90/10 percentile range. *p<0.01, **p<0.001, ***p<0.0001 compared with WT haploid; calculated using a Mann–Whitney U test. (**E**) The bar plot depicts the expression, measured by quantitative RT-PCR, of several normally silenced genes: YFP inserted into the endogenous *HML* locus; subtelomeric genes *YFR057W (Chr III), COS12 (Chr XII), AAD15 (Chr XV)*, and *PAU4 (Chr XII)*; and rDNA

*Figure 1 continued on next page*

*Figure 1 continued*

gene *NTS1-2 (Chr XII)*. Transcriptional levels are plotted as fold expression relative to the WT haploid strain. Error bars represent the standard deviation (SD) of three biological replicates. *p<0.05, **p<0.01, ***p<0.005 compared with WT haploid; calculated using a two-tailed t-test.

DOI: https://doi.org/10.7554/eLife.27991.002

The following video and figure supplement are available for figure 1:

**Figure supplement 1.** Heterogeneous expression pattern of YFP[+] and YFP[-] signals within the aneuploid population was not due to the karyotypic variations.

DOI: https://doi.org/10.7554/eLife.27991.003

**Figure 1—Video 1.** Transitions between repression and derepression of the *HML* locus in proliferating cell lineages with the following karyotypes: Gain of III, III, X.

DOI: https://doi.org/10.7554/eLife.27991.004

**Figure 1—Video 2.** Transitions between repression and derepression of the *HML* locus in proliferating cell lineages with the following karyotypes: Loss of I, V, VII, VIII, XI (basal ploidy, 2N).

DOI: https://doi.org/10.7554/eLife.27991.005

**Figure 1——Video 3.** Transitions between repression and derepression of the *HML* locus in proliferating cell lineages with the following karyotypes: Gain of I, X, XII, XIII.

DOI: https://doi.org/10.7554/eLife.27991.006

(n = 100) obtained through sporulation of a diploid strain carrying the *HML::YFP* reporter as a control (data not shown).

To study how the imbalance of specific chromosomes alters chromatin silencing, we first determined which of the desilenced aneuploid strains had stable karyotypes, since most of the yeast aneuploids obtained through triploid meiosis are karyotypically unstable (*Pavelka et al., 2010*). Using fluorescence-activated cell sorting (FACS) and qPCR karyotyping analyses, as previously described (*Pavelka et al., 2010*), we identified 24 aneuploid strains with unique and stable karyotypes, each with gain or loss of multiple chromosomes compared to the basal ploidy – defined as the copy number possessed by most chromosomes (*Supplementary file 1*). In the desilenced strains, Chromosome (Chr) III, Chr X, Chr XII, and Chr XIII showed the most abundant copy number variation (more specifically, gain) (*Figure 1C*). In particular, Chr III and Chr X were significantly enriched as chromosomes in gained aneuploid numbers in the desilenced aneuploid strains compared with a set of stable aneuploid strains of the same genetic background (S288c) that were isolated through triploid meiosis but not selected for silencing phenotype (*Pavelka et al., 2010*) (*Figure 1C*).

We subsequently focused on the four aneuploid strains showing the most prominent defects in *HML* silencing for more in-depth analysis (*Figure 1B,D*). To exclude the possibility that the silencing defect was caused by spontaneously arising mutations rather than aneuploidy, all four aneuploid strains were subjected to whole-genome sequencing to compare with the parental euploid strains. This analysis revealed an absence of coding region mutations that were not already present in the parental euploid strains (see Materials and methods).

Quantification of the mean fluorescence intensity for each of the four aneuploid populations showed a significant increase in YFP expression compared with the WT haploid control strains (*Figure 1D*). The extent of *HML::YFP* reporter desilencing in the aneuploid strains was comparable to or greater than that of the Δ*sir1* strain (*Figure 1D*). Interestingly, YFP expression was heterogeneous within each aneuploid population, and such heterogeneity was also observed in Δ*sir1* strain as reported previously (*Xu et al., 2006*). To ensure the heterogeneous expression pattern of YFP[+] and YFP[-] signals within the population was not due to the karyotypic variations, cells with 2x Chr III and 1x Chr X gain were sorted into two distinct subpopulations based on YFP fluorescent signal for further analysis, and qPCR karyotyping showed that both sub-populations retained the expected karyotype with gain of Chr III and X (*Figure 1—figure supplement 1B–D*). To determine if this heterogeneity was due to the instability of the chromatin silencing state, we performed time-lapse imaging of YFP expression over several cell divisions in aneuploid strains. We observed transitions between repression and derepression of the *HML* locus in proliferating cell lineages (*Figure 1—figure supplement 1E–G*), which was never observed in haploid lineages (data not shown), suggesting that these aneuploid karyotypes disrupted the stable inheritance of the silenced chromatin.

We next used quantitative RT-PCR (qPCR) analysis to confirm the defective *HML* silencing observed by microscopy. This analysis revealed significant derepression of several genomic loci

across different chromosomes that are normally silenced (*Li et al., 2006*; *Ellahi et al., 2015*), including not only *YFP* at *HML* but also the subtelomeric genes *YFR057W*, *COS12*, *AAD15*, and *PAU4* and the *rDNA* region gene *NTS1-2* (*Figure 1E*). The observation of increased gene expression at all three genomic regions previously implicated in chromatin silencing (*Talbert and Henikoff, 2006*) demonstrates that aneuploidy can affect global gene expression by altering the state of chromatin silencing, in addition to the previously shown effects related directly to changes in DNA copy number (*Torres et al., 2007*; *Rancati et al., 2008*; *Pavelka et al., 2010*; *Sheltzer et al., 2012*).

## Chromosome X disomy is the simplest karyotype that disrupts chromatin silencing

Since each of the above strains had multiple aneuploid chromosomes, we performed segregation analysis to determine which chromosome aneuploidy was linked to the silencing phenotype. We treated two strains, one with gains in Chr III (2x), and X (III, III, X-gain) and the other with gains in Chr I, X, XII, and XIII (I, X, XII, XIII-gain), with low concentrations of radicicol, an inhibitor of Hsp90, to induce chromosomal instability and karyotype changes (*Chen et al., 2012*). We then isolated and karyotyped various aneuploid segregants from these strains (see methods) and analyzed silencing of the *HML::YFP* reporter. Analysis of the aneuploid segregants obtained from the aneuploid strain (III, III, X-gain) showed that the desilencing phenotype co-segregated with Chr X as the only gained chromosome (referred as disome X hereafter) (*Figure 2A*). Interestingly, the gain of one or two extra copies of Chr III, which carries the *HM* loci, did not affect *HML* silencing on its own. Furthermore, qPCR analysis revealed that two extra copies of Chr III did not cause transcriptional derepression of *HML:YFP*, subtelomeric genes, or *rDNA* regions (*Figure 2B*). However, gaining two copies of Chr III did exacerbate the silencing defect of disome X (*Figure 2A*). Disome X segregants independently obtained from the second aneuploid strain (I, X, XII, XIII-gain) also showed significant *HML* desilencing (*Figure 2A*). Conversely, a segregant from the above karyotype with extra copies of Chr I, XII, and XIII, but not Chr X, did not show a significant difference in *HML* silencing compared with the haploid control (*Figure 2A*). These observations establish a causal link between Chr X gain and *HML* desilencing.

To further test whether an acute gain of Chr X would be sufficient to cause *HML* desilencing, we induced Chr X disomy using a previously described conditional centromere strategy (*Reid et al., 2008*; *Anders et al., 2009*). In both WT haploid cells and aneuploid strains with two extra copies of Chr III, we integrated the *GAL1* promoter ($P_{gal1}$) into the region of Chr X directly adjacent to the consensus centromere sequences ($P_{gal1}$-CEN-X). Upon galactose addition, this *GAL1* promoter was activated to induce mitotic non-disjunction of Chr X, resulting in one viable progeny cell with an extra copy of Chr X and an inviable one per cell division. For both the haploid and aneuploid strains that contained the $P_{gal1}$-CEN-X construct, growth in galactose resulted in a significant increase in YFP expression from *HML* compared with corresponding control strains, Chr X::$P_{gal1}$, in which $P_{gal1}$ was integrated into Chr X at sites distant from the consensus centromere sequences (*Figure 2C*). This result further confirmed that gain of Chr X was sufficient to disrupt chromatin silencing at *HML*.

Because the heterogeneity in desilencing was similar between disomy X and Δ*sir1*, we further combined disomy X and Δ*sir1* to test whether these genetic lesions disrupt *HML* silencing through the same or different mechanisms. Notably, the fluorescence of the *HML::YFP* reporter in this double mutant was significantly higher (p<0.05) than in either disome X or Δ*sir1* individually, suggesting an additive or synergistic effect between these genetic abnormalities on *HML* locus silencing (*Figure 2D*).

## Disomy X impairs growth arrest in response to α-factor

*MATa* haploid yeast cells normally respond to the pheromone α-factor by switching from vegetative growth to a G1 cell cycle arrest. Desilencing at *HM* loci results in the expression of both *a*- and α-specific genes and the inability to respond to pheromones (*Osborne et al., 2009*). To test whether the effect of aneuploidy on silencing influences the pheromone-induced growth arrest, we treated WT haploid, Δ*sir1* and disome X strains, all of the *a* mating type with the endogenous copy of the *HML* locus (non-YFP inserted), with α-factor for 90 min. FACS-based cell cycle analysis showed that, while WT haploid cells were fully arrested in the G1 phase, $23 \pm 1.7\%$ of Δ*sir1%* and $14 \pm 1.5\%$ of disome X cells remained in G2 (*Figure 3A*). We next applied filter discs saturated with α-factor to a

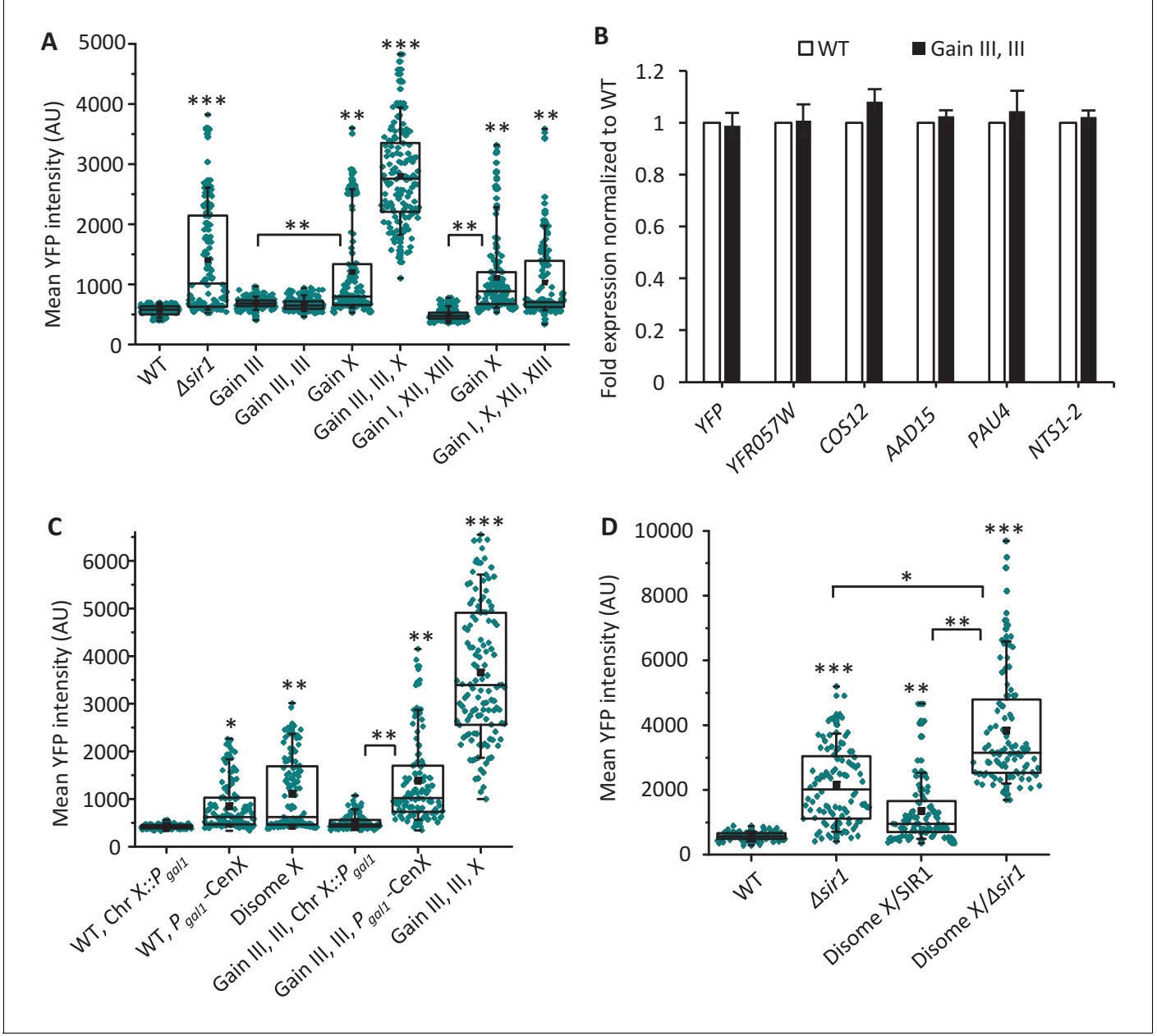

**Figure 2.** Gain of Chr X is sufficient to disrupt silencing. (**A**) The box plot shows mean YFP intensities, determined by microscopy of 125 individual cells for each of the following strains: WT haploid, Δsir1, and two parental aneuploid strains (Gain of III, III, X and Gain of I, X, XII, XIII) and their segregants (Gain of III; Gain of III, III; and Gain of I, XII, XIII). The box spans the first through third quartile values, the line inside each box indicates the median, the solid black square designates the mean, and the whiskers mark the 90/10 percentile range. *p<0.01, **p<0.001, ***p<0.0001 compared with WT haploid unless indicated by brackets; calculated using a Mann–Whitney U test. (**B**) The bar plot depicts the expression, measured by quantitative RT-PCR, of several normally silenced genes in haploid and aneuploid cells with two extra copies of Chr III. These genes are YFP inserted into the endogenous *HML* locus; subtelomeric genes *YFR057W (Chr III), COS12 (Chr XII), AAD15 (Chr XV), and PAU4 (Chr XII); and rDNA gene NTS1-2 (Chr XII)*. Transcription levels are plotted as fold expression relative to the WT haploid strain and not significantly different in Gain III, III strain compared to WT haploid (p<0.05, calculated using a two-tailed t-test). Error bars represent SD of three biological replicates. (**C**) The box plots show mean YFP intensities, determined by microscopy of 125 individual cells for each of the indicated WT haploid or aneuploid strains. The *GAL1* promoter (*Pgal1*) was integrated into Chr X either directly adjacent to (*Pgal1-CEN-X*) or far from (Chr X::*Pgal1*) consensus centromere sequences. The box plot presentation and statistical analysis are performed as described in *Figure 2A*. (**D**) The box plots show mean YFP intensities, determined by microscopy of 125 individual cells for each of the following strains: WT haploid, Δsir1, disome X/*SIR1* and disome X/Δsir1 double mutant. The box plot presentation and statistical analysis are performed as described in *Figure 2A*.

DOI: https://doi.org/10.7554/eLife.27991.007

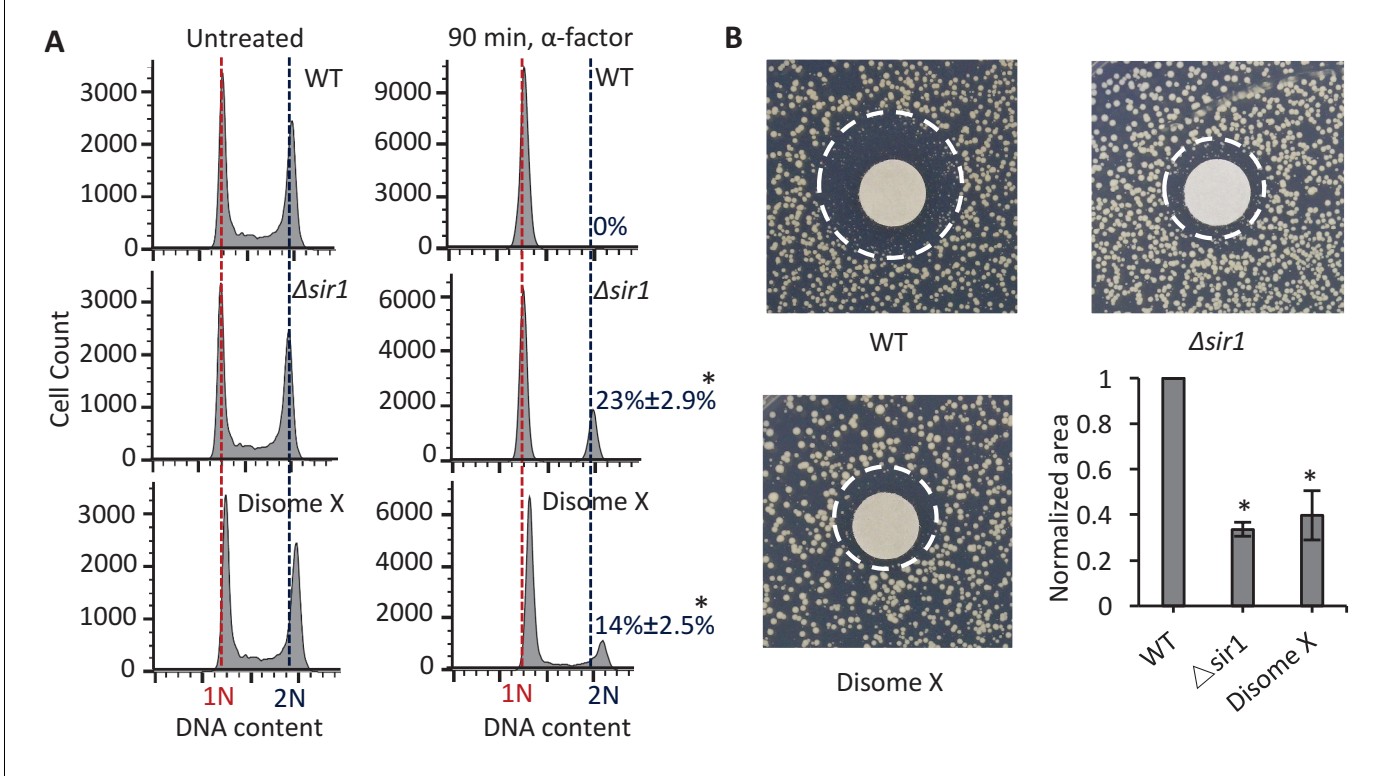

**Figure 3.** Cells with a gain of Chromosome X show abnormal growth arrest in response to α-factor. (**A**) The plots show FACS-based DNA content analysis, indicating cell cycle stage, in *MATa* WT haploid, *Δsir1*, and disome X strains. Left panels represent untreated cells; right panels represent strains treated with 2 μg/ml α-factor for 90 min. Peaks overlapping with the red dotted line represent cells in the G1 phase with a haploid genome content (1N). Peaks overlapping with the blue dotted line represent cells in the G2 phase with a diploid genome content (2N). Percentages are the fraction of total cells in G2,±SD. *p<0.001 compared with WT haploid; calculated using a two-tailed t-test. (**B**) Images depict a pheromone sensitivity assay conducted by applying filter discs carrying α-factor (15 μl of 2 μg/ml) to lawns of WT haploid, *Δsir1*, or disome X *MATa* strains. The images shown were used to calculate the size of the zone devoid of cell growth (the region between the rim of the disc and the dashed circle); these areas, indicative of cellular sensitivity to α-factor, were normalized to the WT haploid strain and plotted. The plot shows the mean and SD from three replicates per strain. *p<0.001 compared with WT haploid; calculated using a two-tailed t-test.

DOI: https://doi.org/10.7554/eLife.27991.008

lawn of WT haploid, *Δsir1* and disome X *MATa* cells. The latter two strains showed significant reductions in pheromone sensitivity compared with the WT haploid population (*Figure 3B*). This demonstrates that the silencing defect of disome X impairs the ability of these aneuploid cells to undergo growth arrest in response to a paracrine factor.

## Chromatin desilencing in disome X is associated with increased acetylation of H4K16 and reduced Sir2 enrichment across *HM* loci

Hypoacetylation of histone H4 at lysine 16 (H4K16) is essential for the establishment and maintenance of silencing at *HM* loci and subtelomeric genes (*Katan-Khaykovich and Struhl, 2005*; *Osborne et al., 2009*). The deacetylation of H4K16 is carried out by Sir2, an NAD-dependent histone deacetylase that localizes to silenced chromatin (*Thurtle and Rine, 2014*). To uncover whether the distribution of acetylated H4K16 (H4K16ac) and occupancy of Sir2 is affected in the disome X strain, we performed chromatin immunoprecipitation (ChIP) using anti-H4K16ac antibody in the haploid, aneuploid, and *Δsir1* strains, followed by qPCR using primer sets spanning *HML* and *HMR*. We observed significantly increased acetylation of H4K16 across both *HM* loci in disome X and *Δsir1* cells compared with WT haploid controls (*Figure 4A–B*). Similarly, we performed ChIP using anti-Sir2::HA antibody and found significantly reduced levels of Sir2 protein localized to both *HM* loci in the disome X and *Δsir1* strains, compared to WT haploid control (*Figure 4C–D*). These results suggest that

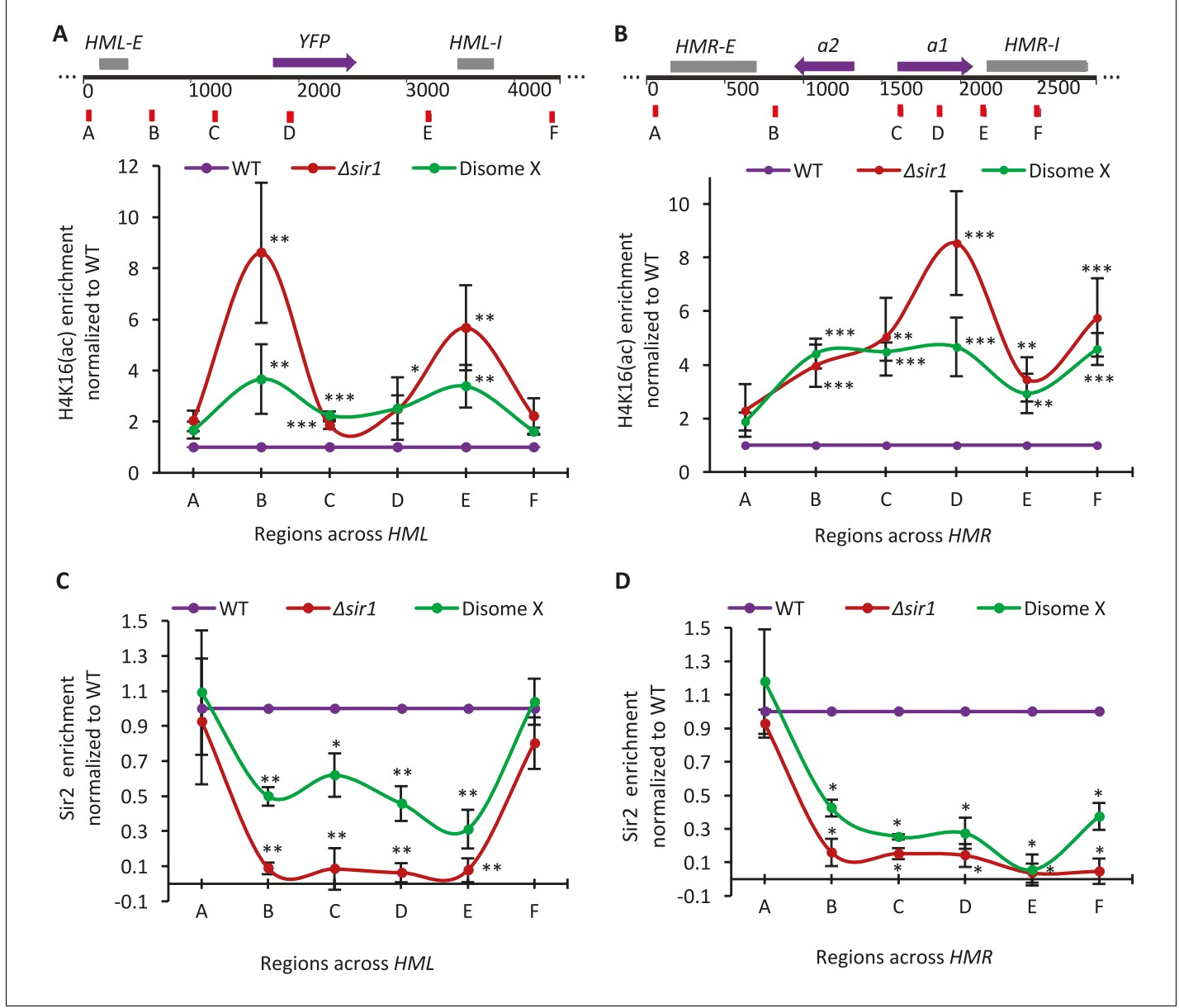

**Figure 4.** *HM* desilencing in disome X cells correlates with increased H4K16 acetylation and reduced Sir2 enrichment across *HM* loci. (**A–B**) Bottom: The plots show levels of H4K16 acetylation across the *HML* (**A**) and *HMR* (**B**) loci in disome X and Δ*sir1* strains relative to WT haploid cells, determined using anti-H4K16ac chromatin immunoprecipitation (ChIP) followed by quantitative RT-PCR (qPCR) analysis. Top: Schematics of the *HM* loci indicate the genomic positioning of primer sets A to F used for qPCR. Plots show the mean and SD from three biological replicates. *p<0.05, **p<0.01, ***p<0.005 calculated using two-tailed t-test and indicate statistically significant difference in H4K16 acetylation level at corresponding genomic locations in Δ*sir1* and disome X strains compared to WT haploid. (**C–D**) The plots indicate Sir2 occupancy across the *HML* (**C**) and *HMR* (**D**) loci in disome X and Δ*sir1* strains relative to WT haploid cells, determined using anti-Sir2::HA ChIP followed by qPCR analysis with the same primer sets depicted in (**A**) and (**B**) for (**C**) and (**D**), respectively. Plots show the mean and SD from three biological replicates. *p<0.01, **p<0.005 compared with WT haploid; calculated using t-test and indicate a statistically significant difference in Sir2 occupancy at corresponding genomic locations across *HM* loci in Δ*sir1* and disome X strains compared to WT haploid.

DOI: https://doi.org/10.7554/eLife.27991.009

the defective silencing of *HM* loci in disome X cells may result from a reduction in chromatin-localized Sir proteins, such as Sir2, and the corresponding increased acetylation of H4K16 at these sites.

# Loss of *HML* silencing in disome X cells is associated with a dispersed distribution of Sir2 and altered chromatin positioning

The silencing of the *HM* loci and subtelomeric regions is associated with characteristic distributions for both Sir2 protein and the chromatin regions within the nucleus (*Andrulis et al., 1998*; *Taddei et al., 2009*). To further understand the mechanism by which disomy X affects chromatin silencing, we examined the localization of endogenous Sir2, tagged at the genomic locus with mTurquoise (mTurq), and observed that the Sir2-mTurq signal was more diffuse in disome X compared with WT haploid cells (*Figure 5A*). We quantified this difference by calculating the coefficient of variation (CV; standard deviation/mean) of fluorescence pixel intensities within the sum projection of each cell (*Figure 5B*). The CV was significantly reduced for Sir2-mTurq fluorescence in disomy X compared with WT haploid populations (p<0.001, two-tailed t-test), particularly in the 35% of aneuploid cells that had a significantly higher (p<0.01) mean expression of YFP from the *HML* locus than the haploid control. The more dispersed Sir2 distribution in disome X cells is consistent with the reduced concentration of Sir2 in silenced chromatin regions (*Figure 4C–D*). This analysis was designed to quantitatively reflect the difference in the 'sharpness' of Sir2 localization while accounting for the difference in overall fluorescence level in each cell (*Figure 5—figure supplement 1A–C*).

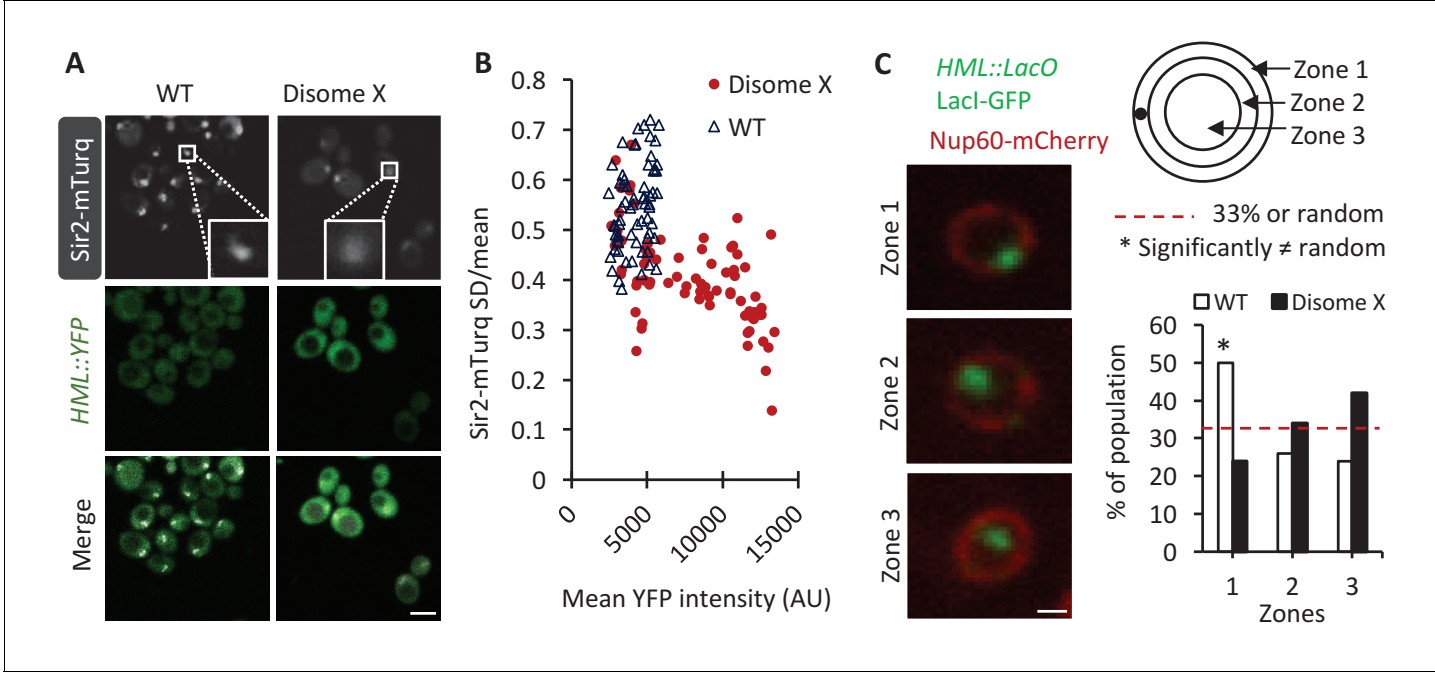

**Figure 5.** Disome X cells display abnormal Sir2 protein localizations and lack proper perinuclear positioning of silenced chromatin region. (A) Representative fluorescent images are shown for Sir2-mTurq and the *HML::YFP* reporter in WT haploid and disome X strains. White boxes in the top panels display magnified images (insets) of representative Sir2 foci. Scale bar, 4 μm. (B) The scatter plots show, for each WT haploid or disome X cell, the coefficient of variation (CV) of Sir2-mTurq fluorescence plotted against the mean YFP pixel intensity. The CV was calculated as the ratio of the standard deviation to the mean pixel intensity of Sir2-mTurq fluorescence over the total area of the sum projection of each cell. The CV in the disome X strain is significantly reduced compared with haploid cells (p<0.001, one-tailed t-test), indicating a more diffusive distribution of Sir2 in aneuploid cells. Additionally, 35% of disome X cells (determined using Tukey's outlier test on the WT strain) have significantly higher (p<0.01) mean YFP intensities than the WT haploid cell population; these disome X cells also show significantly (p<0.05) reduced CV compared with haploid controls. (C) Left: Representative images show the position of the *HML* locus, tagged with *LacO* array and bound by LacI-GFP, relative to the nuclear envelope (NE), marked by Nup60-mCherry. Top right: The illustration shows the three concentric zones of equal area used to map the location of the *HML* locus. Bottom right: The bar graph shows the percentage of WT haploid or disome X cells with GFP puncta located in each of the three zones (n = 100 cells per strain). Confidence values (p) are shown for a χ² analysis comparing random (33% in each zone) and test distributions. *: Value significantly differs from a random distribution (p<0.005, Chi-square test for independence). Scale bar, 1 μm.

DOI: https://doi.org/10.7554/eLife.27991.010

The following figure supplement is available for figure 5:

**Figure supplement 1.** *SIR2* is not haploinsufficient for *HML* silencing

DOI: https://doi.org/10.7554/eLife.27991.011

The positioning of chromatin relative to the nuclear envelope is also important for silencing (*Bystricky et al., 2009*; *Mekhail and Moazed, 2010*). To address whether the normal perinuclear positioning of silenced chromatin was disrupted in disome X cells, we introduced the *LacO* array into the *HML* locus and expressed LacI-GFP to obsereve the *LacO* array-marked site in the nucleus, demarcated with the nuclear envelope (NE) marker Nup60-mCherry. We defined three concentric nuclear zones of equal area - NE (Zone 1), medial (zone 2), and central (zone 3) regions and determined the percentage of cells with the GFP puncta, corresponding to *HML*, in each zone. We observed a significant (p<0.05) reduction in the percentage of disomy X cells with their *HML* locus attached to the NE (zone 1) compared with the WT haploid strain. This was accompanied by a corresponding increase in the percentage of cells with *HML* located in the central nuclear zone (zone 3), suggesting that the silencing defect of the disome X strain is associated with altered chromatin positioning in the nucleus (*Figure 5C*).

## Gain of chromosome X alters subtelomeric gene expression through changes in H3K4me3 and H3K79me3

To determine if disomy X leads to a genome-wide alteration of histone modification, we assessed two histone modifications associated with active chromatin, trimethylation of histone H3 at either Lysine 4 (H3K4me3) or Lysine 79 (H3K79me3), by performing chromatin immunoprecipitation followed by next-generation sequencing (ChIP-seq). In parallel, we analyzed a portion of the same experimental cultures by RNA sequencing (RNA-seq) to correlate genome-wide changes in histone modifications with the transcriptional output. The number of methylated genes (~60% of total *S. cerevisiae* genes) in the disome X and WT haploid strains was not significantly different (*Supplementary file 2*). Consistent with previous reports (*Pokholok et al., 2005*; *Guillemette et al., 2011*; *Takahashi et al., 2011*), both strains showed an overall positive correlation between gene expression and H3K4me3 enrichment, but not H3K79me3 enrichment (*Figure 6—figure supplement 1A*).

To compare the epigenetic and transcriptional changes in aneuploid cells, we plotted the difference in H3K4me3 enrichment between disome X and WT haploid strains against the fold change in gene expression for each of the 3502 genes (~57% of the total genes in the yeast genome, *Supplementary file 2*) that were expressed with detectable H3K4me3 marks in both strains. At the genome-wide level, we found no correlation between an increased enrichment of H3K4me3 modifications and increased gene expression in disome X cells compared with the haploid control (*Figure 6A*). Likewise, there was no difference in gene expression between disome X and haploid populations for sets of genes that had H3K4me3 modifications only in one strain or the other (*Figure 6—figure supplement 1B and C*). However, most of the subtelomeric genes that were expressed and modified in both strains (*Supplementary file 2*) were significantly enriched (p-value=$2.5 \times 10^{-8}$, Fisher's exact test) for H3K4me3 modifications and had higher expression levels in disome X cells compared with the haploid control (*Figure 6A*).

The H3K79me3 histone mark has previously been implicated in the regulation of subtelomeric silencing by modulating the binding of Sir proteins to these chromatin regions. In our ChIP- and RNA- seq experiments, three subtelomeric genes, *COS12, IMD2,* and *YIR042C,* showed significantly increased RNA expression levels and enriched H3K4me3 and H3K79me3 modifications in disome X cells compared with WT (*Figure 6C–D*). Notably, *COS12* was previously identified as a target of H3K79me3-regulated silencing (*Takahashi et al., 2011*). On the genome-wide level or in subsets of genes carrying H3K79me only in one or the other strain, however, we observed no correlation between transcription levels and this epigenetic mark in disome X cells relative to WT (*Figure 6B*, *Figure 6—figure supplement 1D–E* and *Supplementary file 2*).

## The *HML* silencing defect results from increased copy number of at least four genes on chr X

To identify the genetic components that contribute to the silencing defect caused by an extra copy of Chr X, we used the *HML::YFP* reporter and fluorescence microscopy to screen the effect of an increased copy number of each of the 304 genes located on Chr X, for loss of YFP silencing in the disome III background as a means to sensitize the screen. We took advantage of the yeast MOBY library, in which each gene, with its endogenous promoter, is carried on a low-copy (centromeric)

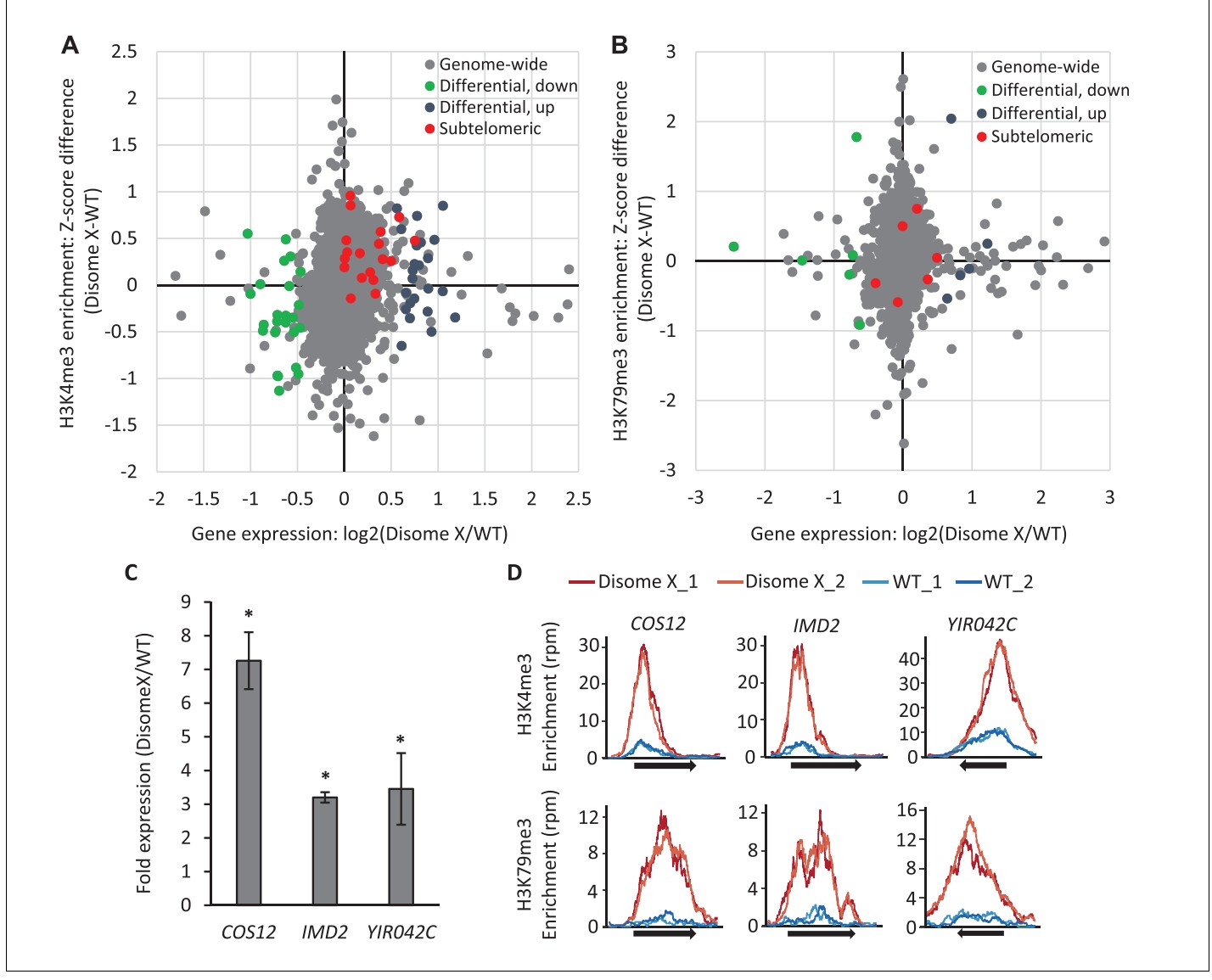

**Figure 6.** Genome-wide analysis shows that disome X cells upregulate histone modifications and transcription of typically silenced genes. (A–B) Gene expression changes determined by RNA-seq are plotted on the X-axis as log2 fold change (disome X/WT haploid), and H3K4me3 (A) and H3K79me3 (B) histone modification enrichments determined by ChIP-seq are plotted on the Y-axis as the difference in Z-scores (disome X - WT haploid), with each dot representing an individual gene. Genes that were expressed (RPKM >1) and enriched for a given histone modification in both haploid and disome X strains were included in this analysis and categorized into four groups: (I) Genome-wide, representing all of the included genes (grey dots); (II and III) Subsets of genes from category I that were significantly (1.5- fold change, p<0.01) upregulated (blue dots) or down-regulated (green dots) in disome X cells compared with haploid controls; and (IV) subtelomeric genes (red dots). Note that identities of subtelomeric genes in A and B are different because the genes with occupancy of the two histone markers (K4me and K79me) were not the same and hence are at different points along the x-axis of A and B. More detailed information about the genes in these categories is listed in *Supplementary file 2*. (C) Transcriptional levels of *COS12* (Chr XII), *IMD2* (Chr VIII), and *YIR042C* (Chr IX) genes were measured by RNA-seq and plotted as fold change (disome X/WT haploid). Each bar depicts the mean and SD of three biological replicates. *p<0.001 compared with WT haploid; calculated using two-tailed t-test. (D) Enrichment profiles of H3K4me3 and H3K79me3 were determined by ChIP-seq and plotted as reads per million per nucleotide (RPM) for the indicated gene ORFs (black arrows), with 500 bp of flanking sequence on both sides. Each plot shows the enrichment profiles for two biological replicates per WT haploid or disome X strain. Both epigenetic marks are enriched at all three genomic loci in disome X cells compared with the haploid controls.

DOI: https://doi.org/10.7554/eLife.27991.012

The following source data and figure supplement are available for figure 6:

**Source data 1.** Source data for genome-wide analysis performed in *Figure 6*.

DOI: https://doi.org/10.7554/eLife.27991.014

*Figure 6 continued on next page*

Figure 6 continued

**Figure supplement 1.** No difference in gene expression between disome X and haploid populations for sets of genes that had H3K4me3 and H3K79me3 modifications only in one strain or the other.

DOI: https://doi.org/10.7554/eLife.27991.013

plasmid (*Ho et al., 2009*). The fifteen Chr X genes with top-ranked silencing defect were further verified by transforming the centromeric plasmid into a WT haploid strain (*Supplementary file 3*). Because cells can maintain up to four copies of centromeric plasmids, we next integrated a single extra copy of each of the top ten candidate genes into a WT haploid genome, but we found that none of these individual genes significantly disrupted silencing when present at this level (data not shown).

Two genes, *RPL39* and *RPS14B*, encoding ribosomal proteins, were among those with the strongest silencing defects when expressed on a centromeric plasmid. Since previous studies showed that overexpression of a ribosomal protein-encoding gene, *RPL32*, impaired silencing (*Singer et al., 1998*), we tested the effect of combining single extra copies, via integration, of *RPL39* and *RPS14B* together in the haploid genome; however, this combination did not alter silencing of the *HML::YFP* reporter (*Figure 7A,B*). Previous studies also showed that silencing is affected by one other top candidates from our MOBY library screen *ASF1*, which encodes a nucleosome assembly factor (*Le et al., 1997*; *Singer et al., 1998*; *Smith et al., 1999*). Integration of *ASF1* together with *DPB11*, a top hit in our screen but not previously known to affect silencing, into the haploid genome also showed no effect on silencing (*Figure 7A,B*). However, when we combined all four genes tested above (*RPL39, RPS14B, ASF1, and DPB11*) by integrating them into the haploid genome, significant *HML* desilencing was observed compared to the haploid control, although the effect was not as

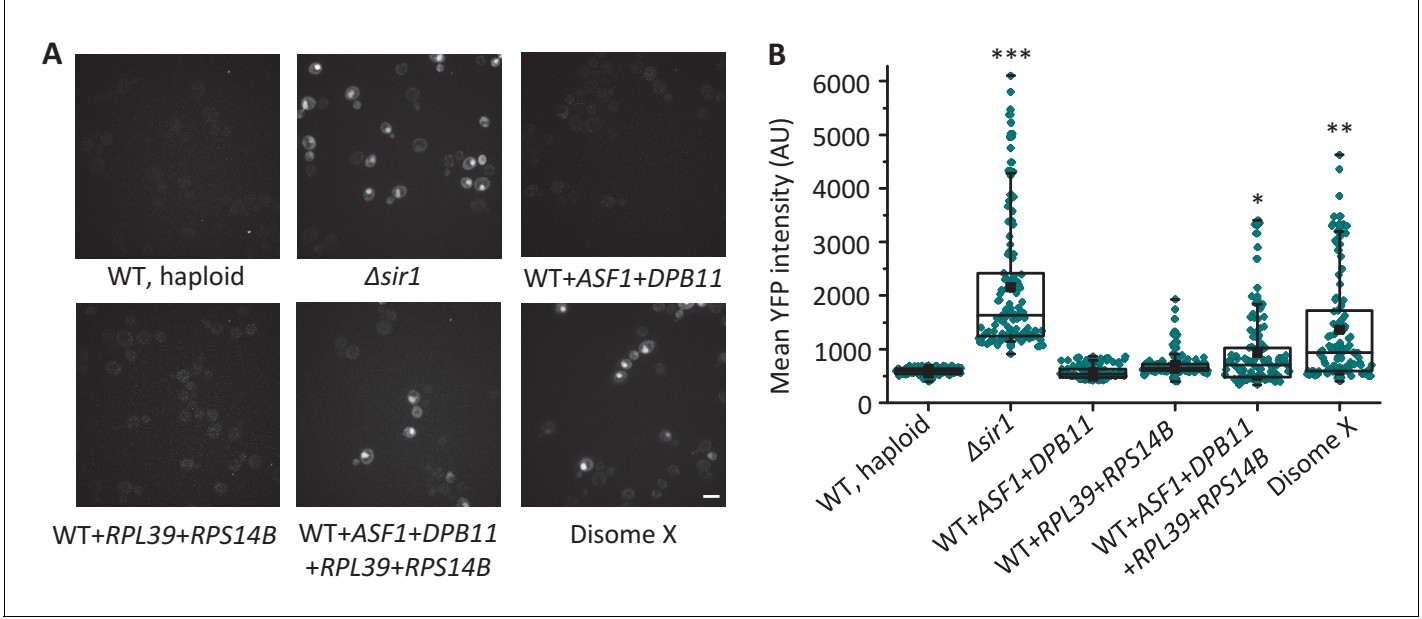

**Figure 7.** The combined increase in copy number of at least four genes on Chr X causes *HML* silencing defects. (**A**) Representative images of YFP fluorescence from the *HML::YFP* reporter in Δ*sir1*, disome X, and WT haploid cells with a single extra copy of the indicated genes, where relevant. Scale bar, 4 μm. (**B**) The box plot shows the mean YFP intensity of 125 cells for each strain shown in *Figure 7A*. The box spans the first through third quartile values, the line inside each box indicates the median, the solid black square designates the mean, and the whiskers mark the 90/10 percentile range. *p<0.01, **p<0.001, ***p<0.0001 compared with WT haploid; calculated using a Mann–Whitney U test.

DOI: https://doi.org/10.7554/eLife.27991.015

The following source data is available for figure 7:

**Source data 1.** Source data for gain-of-funtion screen.

DOI: https://doi.org/10.7554/eLife.27991.016

strong as that in disome X cells (**Figure 7A,B**). These results suggest that the disomy X-induced desilencing is complex and requires the combined effects of at least four Chr X-linked genes.

## Discussion

Our imaging-based analysis of thousands of freshly-produced aneuploid yeast colonies demonstrated that roughly 3% of random aneuploid karyotypes disrupt transcriptional silencing at the *HML* locus, indicating that aneuploidy can impact gene expression to an extent far greater than the effects resulting from direct gene-dosage changes. We identified specific karyotypic features associated with the silencing defect, with the simplest being gain of Chr X, which is sufficient to destabilize the epigenetic state and alter cellular responses to a relevant physiological factor (α-factor). Furthermore, the loss of silencing at the *HM* loci on Chr III and transcriptional derepression at subtelomeric regions on different chromosomes induced by Chr X disomy correlated with changes in the histone modification landscape, including increased H4K16 acetylation, and H3K4 and H3K79 trimethylation. Moreover, the silencing defect of disome X cells was associated with perturbed chromatin localization within the nucleus. The genetic basis of disome X-induced desilencing is complex, requiring at least four Chr X genes. Taken together, our results provide the evidence that aneuploidy can be a direct genetic cause of epigenetic dysregulation.

### Loss of chromatin silencing is associated with specific karyotypic features

Our unbiased approach to identify chromosome stoichiometries associated with disrupted chromatin silencing revealed that Chr III and Chr X were frequently gained in cells lacking stable silencing at the *HML* locus. Notably, this pattern of chromosome enrichment was significantly different from that found in viable, karyotypically-stable aneuploid strains obtained through triploid meiosis (*Pavelka et al., 2010*), suggesting that chromatin desilencing is not an obligatory outcome of abnormal chromosome numbers, but rather, is caused by specific chromosome imbalance. We expected that Chr III and Chr XII would be enriched in our screen since extra copies of these chromosomes would increase the copy number of heterochromatic DNA, which could potentially titrate silencing factors (*Smith et al., 1998*; *Michel et al., 2005*; *Dodson and Rine, 2015*). However, only chromosome III gain was enriched and contributes to desilencing, but it was insufficient on its own for significant desilencing. Chr X, by contrast, lacks any known regions of silenced chromatin other than the subtelomeric DNA, and so the effect of extra copies of this chromosome was unlikely to be related to the titration of silencing factors. Our functional analysis suggests that the mechanisms underlying disome X-associated desilencing are complex (discussed further below).

It is important to point out that the aneuploid karyotypes that we identified did not appear to lead to a single state of chromatin silencing or desilencing but instead resulted in heterogeneous populations of cells with respect to gene expression from the normally silenced chromatin regions. Although the different levels of YFP expression in individual YFP⁺ cells might be due to the natural gene expression noise, the co-existence of considerable fractions of both YFP⁺ and YFP⁻ cells in each of the aneuploid populations suggests that these karyotypes lead to instability of the epigenetic state that is stably inherited in normal haploid cells.

### The impact of chr X disomy on histone modifications at silenced and non-silenced loci

H3K4 trimethylation is considered to be a mark of active transcription, given that its occupancy is generally high at the promoters of actively transcribed genes (*Pokholok et al., 2005*; *Guillemette et al., 2011*). Supporting this notion, our studies show a positive correlation between enrichment of this histone mark and transcriptional activity in both haploid and aneuploid strains. In yeast, H3K4 trimethylation is carried out by the Set1 complex and transcriptional outcomes related to changes in this mark are partially dependent on the location of genes in active or silent chromatin regions (*Bryk et al., 2002*; *Krogan et al., 2002*; *Santos-Rosa et al., 2002*). Although H3K4 is generally hypermethylated within regions of euchromatin and hypomethylated within heterochromatin, loss of H3K4me3 due to deletion of *SET1* had little effect on coding gene expression (*Margaritis et al., 2012*). Consistently, our transcriptome analysis in disome X strains does not show a correlation between H3K4me3 and global transcriptional activation. However, in the case of silent

domains, such as subtelomeric genes, we indeed found that both H3K4me3 modifications and gene expression were increased in disome X cells compared with the haploid control strain. The significant changes in histone modification patterns, combined with transcriptional derepression at many loci in regions of silent chromatin (mating-type loci, *rDNA,* and subtelomeric DNA) in disome X demonstrated that aneuploidy has the capacity to alter the histone-modification profile. It should be noted, however, that RNA-seq data showed that most of the subtelomeric genes are not strongly affected by Chr X disomy, suggesting that this aneuploidy does not have a general effect on silencing at subtelomeres. However, this is consistent with the previous observation that expression of subtelomeric genes in *S. cerevisiae* is largely uninfluenced by mutations in Sir proteins and Sir-based silencing is not a widespread phenomenon at telomeres despite strong enrichment of Sir proteins at telomeric regions (*Takahashi et al., 2011*; *Ellahi et al., 2015*).

## Potential mechanisms by which chr X disomy disrupts stable epigenetic inheritance

We investigated several non-mutually exclusive mechanisms by which gain of Chr X could disrupt stable chromatin silencing. First, the defect in silencing could be due to a moderately reduced Sir2 level in Disome X compared to haploid (*Figure 5—figure supplement 1A–C*). However, this is unlikely because previous study (*Dodson and Rine, 2015*) and our own experiments showed that *SIR2* is not haploinsufficient for silencing: heterozygous *SIR2* diploid strain ($\Delta$sir2/SIR2), in which Sir2 level was reduced by an extent similar to that in disome X, did not compromise *HML* silencing (*Figure 5—figure supplement 1D*). Second, we explored the possibility that NAD biosynthesis could be compromised because three genes (*BNA1, BNA2, BNA4*) involved in this pathway are located on Chr X, and low NAD levels have been linked to defective silencing phenotypes (*Grozinger et al., 2001*; *Sandmeier et al., 2002*; *Bedalov et al., 2003*). However, supplementing disome X strains with NAD did not rescue the desilencing phenotype (data not shown). Third, our comprehensive screen for Chr X genes showed that a combination of four genes partially recapitulated the desilencing phenotype of disome X when each is increased by only a single copy. The known functions of these four genes are diverse, ranging from ribosomal components to a histone chaperone and a DNA polymerase subunit, suggesting that chromatin desilencing in disome X results from the combinatorial effects of multiple pathways that may each contribute to the establishment or maintenance of the silenced chromatin. This is consistent with previous studies showing that aneuploidy confers complex or significant phenotypic changes by multigenic mechanisms (*Selmecki et al., 2006*; *Rancati et al., 2008*; *Selmecki et al., 2008*; *Pavelka et al., 2010*; *Chen et al., 2012*, *2015*).

Another possible mechanism by which aneuploidy could impact silencing is by affecting the defined chromosome organization within the nucleus, whereby heterochromatin-like regions are tethered to the nuclear periphery and form a specialized structural compartment, which is required for Sir proteins to establish silencing (*Andrulis et al., 1998*; *Mekhail et al., 2008*; *Bystricky et al., 2009*; *Ruault et al., 2011*). Indeed, our results show reduced attachment of the *HML* locus to the nuclear envelope in disome X cells. The diffused Sir2 distribution, particularly in cells with desilenced *HML* gene expression, is consistent with previous reports that the silencing function of this protein requires its concentration to perinuclear pools (*Hoppe et al., 2002*; *Taddei et al., 2009*). However, it is presently unclear whether the insufficient tethering of chromosome regions to be silenced to the nuclear periphery or failed concentration of Sir2 to this area of the nucleus is directly caused by the increased copy number of the relevant genes on Chr X. Studies of the transcriptome in trisomy 21 human fibroblasts show that, although the overall nuclear organization defined by lamin-associated domains (LADs) is intact in these cells, alterations in H3K4me3 correlate with specialized gene expression dysregulation domains (GEDDs) (*Letourneau et al., 2014*). This finding together with our results suggests that a perturbed nuclear compartmentalization, which causes changes in gene expression, may be an emergent outcome of gene copy number imbalance associated with certain chromosome aneuploidy. However, our data do not clarify whether the altered chromatin positioning or *HML* perinuclear localization in disome X strain was a cause or consequence of chromatin desilencing.

Epigenetic states are acquired through a precise balance between euchromatin and heterochromatin and are an essential mechanism to control proper cellular identity (*Jaenisch and Bird, 2003*). Here, we have shown that numerical alterations in chromosomes can derepress heterochromatin to break this delicate balance, relax epigenetic inheritance, and cause stochastic variation in cell

identity that impairs the responsiveness to regulatory factors. Our findings provide the causal evidence that aneuploidy is a source of epigenetic instability. It may thus be worth exploring a potential linkage between epigenetic dysregulation and chromosome copy number alterations observed in cancer. The aneuploidy-induced changes in heterochromatin inheritance and histone-modification landscape may be an important mechanism by which chromosomal instability drives large-scale phenotypic variability during tumor evolution.

## Materials and methods

### Yeast strains and plasmids

The yeast strains used in this study were generated in the S288c background and are listed in *Supplementary file 1* and *Supplementary file 4*. To construct the parental strain RLY9017 (*hml::P$_{URA3}$-NLS-YFP*), the RLY2626 strain (*MATa, HML*, S288c background) was crossed with the Y3401 strain (*MATα, hml::P$_{URA3}$-NLS-YFP*, W303 background) generously provided by James Broach (*Xu et al., 2006*). The resulting diploid strain was sporulated to obtain a haploid strain with the genotype *MATα, hml::P$_{URA3}$-NLS-YFP*, which was further backcrossed with RLY2626 five times to get a strain background congenic to S288c. The resulting haploid strain RLY9017 was then converted to a fully isogenic triploid strain carrying *HML::YFP* by cycles of mating-type switching and mating as described in *Figure 1—figure supplement 1A*.

To obtain segregant strains from aneuploid cells, the parental strains were grown in synthetic complete (SC) medium (Sunrise Science Products, Inc., San Diego, CA) containing 25 µg/ml radicicol (Sigma-Aldrich, Saint Louis, MO) for 12 hr at 30°C and then plated on YPD plates at a single-colony density (*Chen et al., 2012*). Single colonies were then selected for further analysis based on karyotype. The RLY9029 and RLY9031 strains were constructed by transforming the *Chr X::P$_{gal1}$-URA$^{KL}$-CenX* cassette, amplified from the DY6304 strain generously provided by Rodney Rothstein using primers WMP5 and WMP6, into the RLY9017 and RLY9024 strains, respectively (*Reid et al., 2008*). The RLY9028 and RLY9030 strains were constructed by transforming the *P$_{gal1}$-URA$^{KL}$* cassette, amplified from the RLB914 strain using primers WMP3 and WMP4, into the RLY9017 and RLY9024 strains. The RLY9033 and RLY9035 strains were obtained by crossing RLY2627 cells (*MATα, HML*) with RLY9025 cells (*MATa, hml::P$_{URA3}$-NLS-YFP;+Chr X*). The resulting trisomy X strain (diploid strain with an extra copy of Chr X) was sporulated and meiotic progenies were selected for genotype (*HML-WT copy*) and karyotype, determined by qPCR, to obtain WT haploid and disome X strains.

The *LacO* array and LacI-GFP fusion protein have been described previously (*Robinett et al., 1996*; *Straight et al., 1996*). Briefly, the RLY9041 and RLY9042 strains carrying insertions of LacI-GFP and *LacO* array 1.5 kb proximal to the *HML* locus were obtained by crossing RLY9035 (*MATa; +Chr X*) cells with the YDB111 (*MATα*) strain generously provided by James Haber (*Bressan et al., 2004*). The resulting trisomy X strain was sporulated, and the dissected meiotic progenies were selected for genotype (*hmlprox::lacO(256)-LEU2; HIS3::P$_{URA3}$-LacI::GFP-KanMX*) by growing on SC-Leu + G418 plates. WT haploid and disome X strains were identified by qPCR-based karyotyping. PCR-mediated homologous recombination was used to C-terminally tag *SIR2* with HA and mTurquoise2, tag *NUP60* with mCherry, and delete *SIR1* by replacing the genomic locus with a *KanMX6* cassette (*Longtine et al., 1998*; *Sheff and Thorn, 2004*); correct integrations were confirmed by PCR-based genotyping.

To construct the plasmid *pWM1 (RLB912)*, ORFs for *RPL39 and RPS14B* were amplified from RLY9017 cells and cloned into *EagI* and *XhoI* sites respectively, into the *pRS306* plasmid. To construct the plasmid *pWM2 (RLB913)*, Gibson assembly was used to clone the indicated ORFs into the *XhoI* site of the *pRS306* plasmid. The RLY9046 and RLY9047 strains were constructed by transforming *NcoI*-digested *pWM1* and *HpaI*-digested *pWM2* into RLY9017 and RLY9018 cells, respectively. To construct the RLY9048 strain, the RLY9046 and RLY9047 strains were crossed and sporulated; the dissected meiotic progenies were selected for the indicated genotypes using a standard PCR-based method. The plasmids used in this study are listed in *Supplementary file 5*.

### Microscopy

To prepare cells for microscopy, yeast strains were grown in SC or drop-out medium for about 18 hr at 25°C before the cultures were diluted to a starting OD$_{600}$ of 0.2 and grown for another five hours

to an $OD_{600}$ of 0.6–0.8. Fluorescence microscopy was performed at room temperature on live cells using a 100× αPlan Fluor NA 1.46 objective on a Zeiss Axiovert 200 M microscope (Zeiss, Jena, Germany), equipped with a Yokogawa CSU-X1 spinning-disk confocal system. Using 488 or 561 nm illumination to excite green or red fluorescent proteins, respectively, a series of optical sections with a step size of 0.5 µm was acquired with a Hamamatsu C9100 EMCCD camera and MetaMorph acquisition software. ImageJ software (v. 1.50e; NIH; RRID:SCR_003070) was used to subtract background, adjust contrast, and generate the final sum projections shown.

Time-lapse imaging was performed on a Perkin Elmer Ultraview VoX system (PerkinElmer, Inc., Waltham, MA) or a Zeiss LSM780 laser scanning confocal microscope (Zeiss, Jena, Germany) with a 63×/1.4 oil Plan-Apochromat objective and Zeiss Definite Focus. To prepare the cells, 10 µl of a mid-log phase culture with an $OD_{600}$ of 0.5 was placed on a thin SC agarose gel pad as described (*Tran et al., 2004*). Z-stack images were acquired with a 0.5 µm step size at 30 min time intervals for 14–18 hr. For each time point, images were adjusted using ImageJ software (v. 1.50e; NIH; RRID: SCR_003070), converted to maximum Z projections, and analyzed for mean fluorescence intensities using Imaris software (Bitplane USA, Concord, MA; RRID:SCR_007370).

## Induction of chromosome non-disjunction using galactose

Strains were grown overnight in SC +2% dextrose medium, diluted 2000 times with SC + 2% raffinose medium (Sunrise Science Products, Inc., San Diego, CA), and grown to saturation at 25°C. Cells were pelleted, washed twice with water, inoculated into SC + 2% raffinose medium, and grown until cultures reached log phase with $OD_{600}$ of 0.6–0.8. Cells were pelleted again, washed twice with water, and grown in SC + 2% galactose medium (Sunrise Science Products, Inc. San Diego, CA) for nine hours (*Anders et al., 2009*). To stop galactose induction, the cells were pelleted, transferred into SC + 2% dextrose medium (Sunrise Science Products, Inc., San Diego, CA), grown for three hours at 25°C, and imaged.

## Selection of stable aneuploid karyotypes

Strains with stable aneuploid karyotypes were selected as previously described (*Pavelka et al., 2010*). Briefly, DNA content was analyzed by fluorescence-activated cell sorting (FACS) for eight randomly picked colonies derived from each of the desired triploid meiosis-generated spores; strains were only selected for further analysis if the DNA content of the eight colonies showed levels of variability similar to those of the wild-type control strain RLY9017, indicating uniform ploidy. For these strains, DNA content was reassessed as before, using cells that were independently revived from frozen stocks three times. Strains that continued to show stable ploidy after repeated rounds of FACS analysis were then karyotyped by qPCR.

## Illumina whole-genome sequencing

Euploid (RLY9017, RLY9019, and RLY9021) and aneuploid (RLY9071, RLY9076, RLY9078, and RLY9079) strains were subjected to whole-genome sequencing. Genomic DNA (gDNA) was extracted from 15 ml of stationary phase yeast cultures, using a standard protocol (*Hoffman, 2001*) with the following modifications. Three consecutive phenol/chloroform/isoamylalcohol extractions, followed by a final chloroform extraction, were performed to reduce protein and phenol contamination, respectively. The gDNA samples were then treated with 50 ng/µl affinity-purified RNase A (Thermo Fisher Scientific, Waltham, MA) for 60 min at 37°C to remove contaminating RNA. Final gDNA yields were quantified with a ND-1000 spectrophotometer (NanoDrop, Thermo Scientific, Waltham, MA). Genomic libraries were prepared according to Illumina's recommendations, except that sonication was used instead of nebulization. Cluster generation and read sequencing were performed according to Illumina's recommendations. 150 bp paired-end reads were collected using the Illumina MiSeq system (Illumina, Inc., San Diego, CA) and aligned to the UCSC sacCer3 reference sequence using the BWA package; RRID:SCR_010910 (*Li and Durbin, 2009*), set at a maximum edit distance of 2 per read and allowing for gapped alignment with a maximum of 5 gap opens and −5 gap extensions. The genome analysis toolkit (https://software.broadinstitute.org/gatk/; RRID:SCR_001876) was used to call variants between the reference genome (sacCer3) and each of the strains. Variants were annotated using SnpEff (http://snpeff.sourceforge.net; RRID:SCR_005191). We then excluded SNPs found across all sequenced strains to eliminate mutations already present in the

euploid background. All potential mutations were then manually inspected in IGV (http://www.broadinstitute.org/igv; RRID:SCR_011793), and SNPs called in repetitive regions, in long poly-A or poly-T stretches, or in regions of low alignment quality were discarded. All remaining SNPs were verified by reanalyzing each strain using Sanger sequencing. This analysis revealed that there were no mutations in coding regions that were not already present in the parental euploid strains. Reads have been deposited in the NCBI Sequence Read Archive (SRA; RRID:SCR_004891) under accession no. SRP105283.

## RNA-seq analysis

Cells were grown in SC medium for 18 hr at 25°C; cultures were then diluted to a starting $OD_{600}$ of 0.2 and grown for five to six hours to an $OD_{600}$ of 0.6. RNA samples were prepared from ten $OD_{600}$ units of the final yeast culture using a standard acid–phenol/chloroform extraction method (*Collart and Oliviero, 2001*), and contaminating gDNA was removed by treating with DNase I (Sigma-Aldrich, Saint Louis, MO). PolyA-selected, 50 bp single-end RNA-seq libraries were prepared using the Illumina TruSeq stranded mRNA sample prep kit (Illumina, Inc., San Diego, CA), quantified using a Bioanalyzer (Agilent, Santa Clara, CA), and sequenced on an Illumina HiSeq 2500 platform. Reads were aligned to the sacCer3 reference genome using Bowtie software (RRID:SCR_005476) with default alignment parameters. Read counts were normalized to chromosome copy number. The resulting binary alignment/map (BAM) files were sorted and indexed using SAM tools (*Li et al., 2009*). Differential gene expression was evaluated using the edgeR library (*Robinson et al., 2010*), and adjusted *p*-values were calculated by the Benjamini-Hochberg procedure. Reads and processed data files have been deposited in NCBI Gene Expression Omnibus (GEO) under accession no. GSE98435.

## Quantitative reverse transcriptase-PCR (qPCR) analysis

RNA was extracted as described above and cDNA was prepared from 2 μg of the resulting total RNA using the Super-Script III reverse transcriptase kit (Thermo Fisher Scientific, Waltham, MA). qPCR was performed using SYBR Green real-time PCR master mix (Quanta Biosciences, Beverly, MA) and analyzed by standard procedures (*Yuan et al., 2006*). Gene expression profiles were normalized to chromosome copy number. The oligos used for qPCR amplification are listed in *Supplementary file 6*.

## ChIP-seq analysis

Chromatin immunoprecipitation was performed as previously described (*Aparicio et al., 2004*). Briefly, yeast cells were grown in 500 ml of SC medium to an $OD_{600}$ of 0.8–0.9 and were cross-linked with 1% formaldehyde (Sigma-Aldrich, Saint Louis, MO) for 20 min before the chromatin was extracted. The chromatin was sonicated using Bioruptor (Diagenode, Denville, NJ) at the high setting for ten cycles of 30 s on/off to yield an average DNA fragment size of 500 bp. Chromatin extracts were diluted in immunoprecipitation (IP) buffer and centrifuged to pellet debris; the resulting supernatant, containing the chromatin solution, was aliquoted for immunoprecipitation as follows. Chromatin was first incubated overnight with antibody at 4°C, then with Dynabeads protein G beads (Thermo Fisher Scientific, Waltham, MA). Beads were washed several times, and DNA was recovered in elution buffer (1% SDS, 0.1 M NaHCO$_3$). Crosslinking was reversed by incubating samples at 65°C overnight, followed by protease treatment, phenol/chloroform extraction, and ethanol precipitation of the recovered DNA. Sequence libraries were constructed and validated using the Illumina library protocol and sequenced using the Illumina HiSeq 2500 system as 50 bp single-end reads. Reads were mapped to the sacCer3 reference genome using Bowtie software (RRID:SCR_005476) with parameters: –best –strata -v2 -m 1. Peaks were called by Model-based Analysis of ChIP-Seq (MACS2) using default settings (*Zhang et al., 2008*), mapped to the closest gene, and kept only if they occurred within 600 bases of the transcription start site. Peak scores, defined as the -log10 transformed q-values, normalized to chromosome copy number, were converted to Z-scores for comparison across strains and plotted as the difference between disome X and WT haploid strains (*Figure 6A–B*). Antibodies used for immunoprecipitation are: anti-HA (12CA5, Sigma-Aldrich, Saint Louis, MO; RRID:AB_514505), anti-H3K79me3 (ab2621, Abcam, Cambridge, MA; RRID:AB_303215), anti-H3K4me3 (04–745, EMD Millipore, Temecula, California; RRID:AB_1163444), and anti-

H4K16ac (07–329, EMD Millipore, Temecula, California; RRID:AB_310525). Reads and processed data files have been deposited in NCBI Gene Expression Omnibus (GEO; RRID:SCR_005012) under accession no. GSE98282.

## Pheromone sensitivity assay

For cell cycle analysis, $10^7$ mid-log phase cells were grown in SC medium containing 2 µg/ml α-factor (US Biological, Salem, MA) for 90 mins at 30°C. Cells were then fixed in 70% ethanol and analyzed for DNA content using an Attune NxT flow cytometer (Thermo Fisher Scientific, Waltham, MA) as described (*Pavelka et al., 2010*). For the halo assay, 15 µl of 2 µg/ml α-factor was applied to filter discs centered on a lawn of *MATa* cells with a WT *HML* locus. Cells were grown for 2–3 days at 30°C, and the size of the halo (region devoid of cell growth) was determined as described previously (*Cherkasova et al., 1999*).

## Gain-of-function screen

304 of the 356 total Chr X ORFs are available in the Molecular Barcoded Yeast (MoBY) ORF plasmid library (*Ho et al., 2009*) and used for the screen. Each plasmid was extracted and transformed independently into the disome III strain RLY9023 ($HML::P_{URA3}$-YFP) as described previously (*Chen et al., 2015*). Transformants were grown in 96-well deep-well blocks containing 2 ml SC-Ura medium at 30°C for 12 hr. Cells were then fixed with 1% paraformaldehyde and imaged on the Operetta high-content imaging system (PerkinElmer, Inc., Waltham, MA) with a 63×/1.4 Plan-Apochromat objective. Desilencing scores were calculated as the ratio of the mean YFP fluorescence intensity of a test strain carrying a MoBY plasmid to that of the disome X strain carrying an empty MoBY vector (RLY9046).

## Acknowledgements

We thank James Broach, James E Haber and Rodney Rothstein for providing yeast strains; Adam Rudner for providing Sir2 antibody; Guangbo Chen, Tamara Potapova, Andrei Kucharavy, and Swaminathan Venkatesh, and Andy Feinberg for helpful advice; Kristen Swaney for critically reading the manuscript; Jay Unruh, Brian Slaughter, Sreekumar Ramachandran, Neil Neumann, and Dan Georgess for assistance on microscopy and image analysis; W McDowell and Brian Flaherty for help with qPCR; and Akshay Narkar for help with western blotting. This work was supported by NIH grant R35-GM118172 to RL, American Heart Association predoctoral fellowship 15PRE25090204 to WM, and Prostate Cancer Foundation Young Investigator Award 16YOUN21 to HJT. The funders had no role in study design, data collection, and interpretation, or the decision to submit the work for publication.

## Additional information

### Funding

| Funder | Grant reference number | Author |
| --- | --- | --- |
| National Institute of General Medical Sciences | R35-GM118172 | Rong Li |
| American Heart Association | 15PRE25090204 | Wahid A Mulla |
| Prostate Cancer Foundation | 16YOUN21 | Hung-Ji Tsai |

The funders had no role in study design, data collection and interpretation, or the decision to submit the work for publication.

### Author contributions

Wahid A Mulla, Conceptualization, Data curation, Formal analysis, Funding acquisition, Validation, Investigation, Visualization, Methodology, Writing—original draft, Writing—review and editing; Chris W Seidel, Software, Formal analysis, Investigation, Methodology; Jin Zhu, Scott McCroskey, Juliana Conkright, Allison Peak, Kathryn E Malanowski, Investigation, Methodology; Hung-Ji Tsai, Resources,

Investigation, Writing—review and editing; Sarah E Smith, Software, Formal analysis, Methodology; Pushpendra Singh, Formal analysis, Investigation; William D Bradford, Resources, Investigation, Methodology; Anjali R Nelliat, Formal analysis, Methodology; Anoja G Perera, Methodology; Rong Li, Conceptualization, Supervision, Funding acquisition, Writing—review and editing

**Author ORCIDs**
Wahid A Mulla (iD) http://orcid.org/0000-0003-3356-3902
Rong Li (iD) http://orcid.org/0000-0002-0540-6566

**Decision letter and Author response**
Decision letter https://doi.org/10.7554/eLife.27991.030
Author response https://doi.org/10.7554/eLife.27991.031

## Additional files

### Supplementary files
• Supplementary file 1. The karyotypes of stable aneuploid strains that exhibit defective silencing of YFP at the *HML* locus obtained from a microscopy-based screen are listed.
DOI: https://doi.org/10.7554/eLife.27991.017

• Supplementary file 2. The number of genes plotted for each category in *Figure 6A–B* and *Figure 6—figure supplement 1* is listed.
DOI: https://doi.org/10.7554/eLife.27991.018

• Supplementary file 3. A list of fifteen Chr X genes that cause the strongest silencing defects as a result of increased copy number. Genes leading to the loss of *HML::YFP* silencing when copy number is increased are listed with a functional description and a desilencing score. The desilencing score was calculated as the average YFP intensity in WT haploid strains carrying individual candidate genes on a low-copy (centromeric) plasmid, relative to the average YFP fluorescence in the disome X strain. Average YFP intensities were calculated using three biological replicates per strain.
DOI: https://doi.org/10.7554/eLife.27991.019

• Supplementary file 4. List of yeast strains used in this study, and not listed in *Supplementary file 1*.
DOI: https://doi.org/10.7554/eLife.27991.020

• Supplementary file 5. List of plasmids used in this study.
DOI: https://doi.org/10.7554/eLife.27991.021

• Supplementary file 6. List of oligos used in this study.
DOI: https://doi.org/10.7554/eLife.27991.022

• Transparent reporting form
DOI: https://doi.org/10.7554/eLife.27991.023

### Major datasets
The following datasets were generated:

| Author(s) | Year | Dataset title | Dataset URL | Database, license, and accessibility information |
|---|---|---|---|---|
| Mulla W | 2017 | DNA-Seq analysis of aneuploid strains | https://www.ncbi.nlm.nih.gov/geo/query/acc.cgi?acc=GSE98282 | Publicly available at the NCBI Gene Expression Omnibus (accession no:SRP105283) |
| Mulla W | 2017 | ChIP Seq analysis of H3K4me3 and H3K79me3 in euploid and aneuploid Saccharomyces cerevisiae | https://www.ncbi.nlm.nih.gov/geo/query/acc.cgi?acc=GSE98282 | Publicly available at the NCBI Gene Expression Omnibus (accession no: GSE98282) |

| Mulla W | 2017 | RNA Seq analysis of WT haploid and Disome 10 in Saccharomyces cerevisiae | http://www.ncbi.nlm.nih.gov/geo/query/acc.cgi?acc=GSE98435 | Publicly available at the NCBI Gene Expression Omnibus (accession no: GSE98435) |

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
