## [Decision Letter]

Thank you for submitting your article "Aneuploidy as a cause of epigenetic instability and cell identity disorder" for consideration by *eLife*. Your article has been reviewed by two peer reviewers, and the evaluation has been overseen by a Reviewing Editor and Jessica Tyler as the Senior Editor. The following individual involved in review of your submission has agreed to reveal his identity: Marc Gartenberg (Reviewer #2).

The reviewers have discussed the reviews with one another and the Reviewing Editor has drafted this decision to help you prepare a revised submission.

General assessment:

This manuscript describes an interesting approach to investigate the hypothesis that aneuploidies affect chromatin status and identified specific aneuploid chromosomes and genes on those chromosomes that do so. The authors find that alterations in chromosome dosage that have the potential to disrupt gene silencing in a model organism and suggest that this resembles events that occur during the development of cancer cells. Previous studies have shown that high copies of some genes affect telomere length and/or telomeric silencing. For example, the first telomerase RNA gene was identified in a classic screen for high-copy disruptors of telomeric heterochromatin (Singer Gottschling, 1994). Nonetheless, the observation that a single extra copy of several genes on a single chromosome can alter chromatin-mediated silencing is novel. The work is thorough and represents an interesting conceptual advance, albeit one that might have been expected by the earlier high-copy disruptor studies mentioned above.

Central conclusions:

1) In a screen of isolates carrying randomly generated aneuploidies, 3% were accompanied by derepression of all three major heterochromatin domains: the silenced mating-type loci, sub-telomeric regions and the rDNA.

2) An extra copy of a single chromosome X was sufficient to derepress silencing at the mating locus and this was attributed to four genes on that chromosome.

3) This aneuploidy caused loss of several characteristic features of heterochromatin: a) H4K16ac, H3K4me and H3K79me levels were elevated,b) one of the heterochromatin proteins (Sir2) was mislocalized, andc) heterochromatin domains were no longer perinuclear.

4) The aneuploid strains yielded semi-heritable defects, due to mixed populations of expression states.

The work is interesting, but a bit over-sold and the connection to cancer is rather general. Given that it was known that overexpression of some genes influence silencing, and since chromosome amplification is one mechanism of gene over-expression, the interesting advance is that only specific chromosomes and specific genes on those chromosomes have an effect when present in a single extra copy.

How most of the genes identified contribute to silencing is not clear. Furthermore, whether the defects in heterochromatin were the cause or consequence of the silencing phenotype was left unresolved.

The title of the manuscript should be modified and "cell identity disorder" should be removed. We think the emphasis is better placed on the idea that aneuploidy is a single-step event that can over-express a suite of genes that affect MAT silencing.

Essential revisions:

The reviewers raise a number of important concerns that must be adequately addressed before the paper can be accepted. Some of the required revisions will likely require further experimentation within the framework of the presented studies and techniques, but they should be quite straight-forward.

1) For all of the RT-qPCR experiments, the authors plot the standard error of the mean, which is not appropriate – the standard deviation must be shown and the data may not be statistically significant if the SEM is that big. The authors cannot say that silencing [at loci beyond the MAT locus] is affected based on the current data analysis. Other qPCR figures, including Figure 2—figure supplement 1, should also plot the average and standard deviation instead of SEM. Also, in Figure 6 the authors plot mean and SEM instead of the appropriate SD.

2) In the RNA-seq it doesn't appear that silenced genes come up as differentially expressed (certainly most of the subtelomeric genes aren't strongly affected). Thus, it appears that this aneuploidy does not have an obvious effect on silencing at the telomeres. This should be acknowledged.

3) The heterogeneity in YFP/silencing recapitulates the *sir1* effect, but given the instability of chromosome amplification cited, the authors should show that the heterogeneity in silencing is not due to heterogeneity in chromosome content in the aneuploid strains. This can be performed by picking individual colonies or FACS sorting based on YFP fluorescence and measuring DNA content for the extra chromosomes (by qPCR for the affected chromosome).

4) Subsection “Chromosome X disomy alone alters chromatin silencing”, last paragraph: – We suggest removing Figure 2 as it does not go into sufficient depth and has raised concerns because: 1) the half-life of YFP maturation is 40 minutes (Gordon Brent 2007) and the obligatory lag between derepression and detection could mean that derepression occurs earlier than bud emergence; 2) only 9 cells were examined; 3) the threshold at which YFP can be detected should be determined (It is clearly present at 30 minutes in Figure 2 but not recorded quantitatively until 60 minutes in Figure 2); and 4), there is neither definitive evidence that silencing is transiently lost during passage through S phase nor is there a strict requirement for S phase in silencing establishment (see the review by Gartenberg Smith 2016).

5) Figure 6: if the x-axis is the same in both plots, then why are the red 'subtelomeric' genes at different points along the x-axis? Is the data in this figure significant? This should be stated in the text and clearly shown with mean and SD in the figures.

6) Subsection “Potential mechanisms by which Chr X disomy disrupts stable epigenetic inheritance”, second paragraph – - concentrated Sir proteins and perinuclear localization of heterochromatin are read-outs for intact heterochromatin and loss of either might impair heterochromatin. However, their loss might simply accompany the disruption of heterochromatin by some other mechanism (i.e. they are not necessarily causative events). Improper assembly of chromatin genome-wide can affect heterochromatin function by an anti-silencing mechanism (Gartenberg Smith 2016). This is an additional possibility that should be considered, particularly given that genome-wide acting factors like ASF1 and DPB11 were included in the set of chromosome X genes.

---

## [Author Response]

[…] The title of the manuscript should be modified and "cell identity disorder" should be removed. We think the emphasis is better placed on the idea that aneuploidy is a single-step event that can over-express a suite of genes that affect MAT silencing.

We appreciate the reviewers’ suggestion about the title of the manuscript. We have modified the title as “Aneuploidy as a cause of impaired chromatin silencing and mating-type specification in budding yeast”. We reason that this accurately summarizes the main point of the paper. Because we have observed that aneuploidy with chromosome loss could also affect silencing, we cannot generally conclude gene overexpression underlies the aneuploidy’s effect. Also, our data do not rule out that some aneuploidy could affect silencing more directly by altering nuclear organization than by affecting the expression of specific genes.

Essential revisions:The reviewers raise a number of important concerns that must be adequately addressed before the paper can be accepted. Some of the required revisions will likely require further experimentation within the framework of the presented studies and techniques, but they should be quite straight-forward.1) For all of the RT-qPCR experiments, the authors plot the standard error of the mean, which is not appropriate – the standard deviation must be shown and the data may not be statistically significant if the SEM is that big. The authors cannot say that silencing [at loci beyond the MAT locus] is affected based on the current data analysis. Other qPCR figures, including Figure 2—figure supplement 1, should also plot the average and standard deviation instead of SEM. Also, in Figure 6 the authors plot mean and SEM instead of the appropriate SD.

We thank reviewers for their helpful comment on the statistical analysis. For all the RT-qPCR experiments, the standard deviation is now shown per reviewers’ suggestion. However, this does not change statistically significance and *p-*values are clearly stated in figure legends corresponding to the relevant figures. Our statistical analysis associated with Figure 1 shows that silencing at loci beyond the MAT locus was significantly affected.

2) In the RNA-seq it doesn't appear that silenced genes come up as differentially expressed (certainly most of the subtelomeric genes aren't strongly affected). Thus, it appears that this aneuploidy does not have an obvious effect on silencing at the telomeres. This should be acknowledged.

We appreciate reviewers’ comment and have acknowledged that based on RNAseq data most of the subtelomeric genes were not strongly affected, suggesting that disome X did not have a general effect on silencing at the telomeres, even though some telomeric genes were altered both in expression and in their H3K4 methylation status in the disome X strain (Figure 6). We point out that this is consistent with the previous observation that expression of subtelomeric genes in *S. cerevisiae* is largely uninfluenced by Sir proteins and Sir-based silencing was not a widespread phenomenon at telomeres despite strong enrichment of Sir proteins at telomeric regions (Takahashi, Schulze et al. 2011, Ellahi, Thurtle et al. 2015) (subsection “The impact of Chr X disomy on histone modifications at silenced and non-silenced loci”).

3) The heterogeneity in YFP/silencing recapitulates the sir1 effect, but given the instability of chromosome amplification cited, the authors should show that the heterogeneity in silencing is not due to heterogeneity in chromosome content in the aneuploid strains. This can be performed by picking individual colonies or FACS sorting based on YFP fluorescence and measuring DNA content for the extra chromosomes (by qPCR for the affected chromosome).

We thank reviewers for pointing this out and we were fully aware of the potential karyotype instability of aneuploid strains in our analysis. Because of this awareness, aneuploid strains used in the current study were carefully chosen for their karyotypic stability and homogeneity (explained in the second paragraph of the subsection “Diverse aneuploid karyotypes can cause chromatin desilencing”, and Figure 1). Briefly, DNA content was analyzed by fluorescence-activated cell sorting (FACS) for eight randomly picked colonies derived from each of the original spore colonies showing silencing defect. Strains were only selected for further analysis if the DNA contents of the eight colonies were identical and the FACS profile showed a level of variability (CV) similar to that of the wild-type haploid control strain, consistent with uniform ploidy. For these strains, DNA content was reassessed in cultures that were revived from frozen stocks and this was repeated three times. Strains that continued to show stable and homogeneous ploidy after repeated rounds of FACS analysis were then karyotyped by qPCR, where three colonies from the same plate were analyzed separately to show consistent karyotypes. Moreover, karyotypes were tested before and after every single experiment and no deviation was observed. This procedure for isolating inherently stable aneuploid strains was used in our previous study (Pavelka, Rancati et al. 2010). In our 2012 paper (Zhu et al. 2012) where karyotype instability was studied in-depth, we showed that not all aneuploid strains exhibit karyotypes instability, and some can be quite stable. Because YFP^-^ andYFP^+^ cells represented considerable portions of each population, if these correspond to different karyotypes, our FACS and karyotyping assays would not have shown a distinct ploidy and karyotype, respectively. Furthermore, if YFP^+^ signal in each population had resulted from random karyotype changes due to karyotype instability, the frequency of desilencing karyotypes would have been much higher than the observed 3% in our initial screen.

We did attempt to FACS sort as reviewers suggested, but sorting only worked for the strain with Chr III, III, X gain, which had the highest YFP^+^ population. Karyotyping of YFP^+^ and YFP^-^ cells showed that both retained the expected karyotype. This result has been added to Figure 1—figure supplement 1 and described in the third paragraph of the subsection “Diverse aneuploid karyotypes can cause chromatin desilencing”. FACS sorting for the other aneuploid strains did not work out, due to two reasons: 1) YFP signal in these strains was overall weaker; and 2) not enough cells with fluorescence sufficiently high for sorting could be obtained for qPCR karyotyping.

4) Subsection “Chromosome X disomy alone alters chromatin silencing”, last paragraph: – We suggest removing Figure 2 as it does not go into sufficient depth and has raised concerns because: 1) the half-life of YFP maturation is 40 minutes (Gordon Brent 2007) and the obligatory lag between derepression and detection could mean that derepression occurs earlier than bud emergence; 2) only 9 cells were examined; 3) the threshold at which YFP can be detected should be determined (It is clearly present at 30 minutes in Figure 2 but not recorded quantitatively until 60 minutes in Figure 2); and 4), there is neither definitive evidence that silencing is transiently lost during passage through S phase nor is there a strict requirement for S phase in silencing establishment (see the review by Gartenberg Smith 2016).

We have removed Figure 2 as reviewers suggested.

5) Figure 6: if the x-axis is the same in both plots, then why are the red 'subtelomeric' genes at different points along the x-axis? Is the data in this figure significant? This should be stated in the text and clearly shown with mean and SD in the figures.

We apologize for not explaining this figure clearly and appreciate the reviewers’ comment. Identities of subtelomeric genes in Figure 6 were different because the genes with occupancy of each of the histone markers (K4me and K79me) were not the same (see Table 2) and hence locate to different points along the x-axis of A and B. We have added this explanation in the legend of Figure 6. We performed Fisher’s exact test to compare two groups- subtelomeric (red) and genome-wide (gray) genes falling in the first quadrant in Figure 6, vs. the other three quadrants. The difference is significant, and test statistics are now clearly stated in the second paragraph of the subsection “Gain of Chromosome X alters subtelomeric gene expression through changes in H3K4me3 and 3K79me3”.

6) Subsection “Potential mechanisms by which Chr X disomy disrupts stable epigenetic inheritance”, second paragraph – concentrated Sir proteins and perinuclear localization of heterochromatin are read-outs for intact heterochromatin and loss of either might impair heterochromatin. However, their loss might simply accompany the disruption of heterochromatin by some other mechanism (i.e. they are not necessarily causative events). Improper assembly of chromatin genome-wide can affect heterochromatin function by an anti-silencing mechanism (Gartenberg Smith 2016). This is an additional possibility that should be considered, particularly given that genome-wide acting factors like ASF1 and DPB11 were included in the set of chromosome X genes.

We fully agree with reviewers that it is unclear whether loss of Sir2 concentration and *HML* perinuclear localization were the cause or consequence of desilencing. We, in fact, did not want to imply that they were the cause. We have made a clearer discussion about this in the second paragraph of the subsection “Potential mechanisms by which Chr X disomy disrupts stable epigenetic inheritance”.